# GraphAF: a Flow-based Autoregressive Model for Molecular Graph Generation

**Chence Shi\*[1], Minkai Xu\*[2], Zhaocheng Zhu[3,4], Weinan Zhang[2], Ming Zhang[1], Jian Tang[3,5,6]**

[1]Department of Computer Science, Peking University, China
[2]Shanghai Jiao Tong University, China
[3]Mila - Québec AI Institute, Canada
[4]Université de Montréal, Canada
[5]HEC Montréal, Canada
[6]CIFAR AI Research Chair

`{chenceshi,mzhang_cs}@pku.edu.cn`
`{mkxu,wnzhang}@apex.sjtu.edu.cn`
`zhaocheng.zhu@umontreal.ca`
`jian.tang@hec.ca`

## Abstract

Molecular graph generation is a fundamental problem for drug discovery and has been attracting growing attention. The problem is challenging since it requires not only generating chemically valid molecular structures but also optimizing their chemical properties in the meantime. Inspired by the recent progress in deep generative models, in this paper we propose a flow-based autoregressive model for graph generation called GraphAF. GraphAF combines the advantages of both autoregressive and flow-based approaches and enjoys: (1) high model flexibility for data density estimation; (2) efficient parallel computation for training; (3) an iterative sampling process, which allows leveraging chemical domain knowledge for valency checking. Experimental results show that GraphAF is able to generate 68% chemically valid molecules even without chemical knowledge rules and 100% valid molecules with chemical rules. The training process of GraphAF is two times faster than the existing state-of-the-art approach GCPN. After fine-tuning the model for goal-directed property optimization with reinforcement learning, GraphAF achieves state-of-the-art performance on both chemical property optimization and constrained property optimization.[1]

## 1 Introduction

Designing novel molecular structures with desired properties is a fundamental problem in a variety of applications such as drug discovery and material science. The problem is very challenging, since the chemical space is discrete by nature, and the entire search space is huge, which is believed to be as large as $10^{33}$ (Polishchuk et al., 2013). Machine learning techniques have seen a big opportunity in molecular design thanks to the large amount of data in these domains. Recently, there are increasing efforts in developing machine learning algorithms that can automatically generate chemically valid molecular structures and meanwhile optimize their properties.

Specifically, significant progress has been achieved by representing molecular structures as graphs and generating graph structures with deep generative models, *e.g.*, Variational Autoencoders (VAEs) (Kingma & Welling, 2013), Generative Adversarial Networks (GANs) (Goodfellow et al., 2014) and Autoregressive Models (Van Oord et al., 2016). For example, Jin et al. (2018) proposed a Junction Tree VAE (JT-VAE) for molecular structure encoding and decoding. De Cao & Kipf (2018) studied how to use GANs for molecular graph generation. You et al. (2018a) proposed an approach called Graph Convolutional Policy Network (GCPN), which formulated molecular graph generation as a sequential decision process and dynamically generates the nodes and edges based on the

---

*Equal contribution, with order determined by flipping a coin. Work was done during internship at Mila.

[1]Code is available at https://github.com/DeepGraphLearning/GraphAF

Table 1: Previous state-of-the-art algorithms for molecular graph generation. The comparison of training is only conducted between autoregressive models.

| Name | Generative Model | | | | Sampling Process | | Training Process | |
|---|---|---|---|---|---|---|---|---|
| | VAE | GAN | RNN | Flow | One-shot | Iterative | Sequential | Parallel |
| JT-VAE | ✓ | - | - | - | - | ✓ | - | - |
| RVAE | ✓ | - | - | - | ✓ | - | - | - |
| GCPN | - | ✓ | - | - | - | ✓ | ✓ | - |
| MRNN | - | - | ✓ | - | - | ✓ | ✓ | - |
| GraphNVP | - | - | - | ✓ | ✓ | - | - | - |
| GraphAF | - | - | - | ✓ | - | ✓ | - | ✓ |

existing graph substructures. They used reinforcement learning to optimize the properties of generated graph structures. Recently, another very related work called MolecularRNN (MRNN) (Popova et al., 2019) proposed to use an autoregressive model for molecular graph generation. The autoregressive based approaches including both GCPN and MRNN have demonstrated very competitive performance in a variety of tasks on molecular graph generation.

Recently, besides the aforementioned three types of generative models, normalizing flows have made significant progress and have been successfully applied to a variety of tasks including density estimation (Dinh et al., 2016; Papamakarios et al., 2017), variational inference (Kingma et al., 2016; Louizos & Welling, 2017; Rezende & Mohamed, 2015), and image generation (Kingma & Dhariwal, 2018). Flow-based approaches define invertible transformations between a latent base distribution (*e.g.* Gaussian distribution) and real-world high-dimensional data (*e.g.* images and speech). Such an invertible mapping allows the calculation of the exact data likelihood. Meanwhile, by using multiple layers of non-linear transformation between the hidden space and observation space, flows have a high capacity to model the data density. Moreover, different architectures can be designed to promote fast training (Papamakarios et al., 2017) or fast sampling (Kingma et al., 2016) depending on the requirement of different applications.

Inspired by existing work on autoregressive models and recent progress of deep generative models with normalizing flows, we propose a flow-based autoregressive model called GraphAF for molecular graph generation. GraphAF effectively combines the advantages of autoregressive and flow-based approaches. It has a high model capacity and hence is capable of modeling the density of real-world molecule data. The sampling process of GraphAF is designed as an autoregressive model, which dynamically generates the nodes and edges based on existing sub-graph structures. Similar to existing models such as GCPN and MRNN, such a sequential generation process allows leveraging chemical domain knowledge and valency checking in each generation step, which guarantees the validity of generated molecular structures. Meanwhile, different from GCPN and MRNN as an autoregressive model during training, GraphAF defines a feedforward neural network from molecular graph structures to the base distribution and is therefore able to compute the exact data likelihood in parallel. As a result, the training process of GraphAF is very efficient.

We conduct extensive experiments on the standard ZINC (Irwin et al., 2012) dataset. Results show that the training of GraphAF is significantly efficient, which is two times faster than the state-of-the-art model GCPN. The generated molecules are 100% valid by incorporating the chemical rules during generation. We are also surprised to find that even without using the chemical rules for valency checking during generation, the percentage of valid molecules generated by GraphAF can be still as high as 68%, which is significantly higher than existing state-of-the-art GCPN. This shows that GraphAF indeed has the high model capability to learn the data distribution of molecule structures. We further fine-tune the generation process with reinforcement learning to optimize the chemical properties of generated molecules. Results show that GraphAF significantly outperforms previous state-of-the-art GCPN on both property optimization and constrained property optimization tasks.

## 2 RELATED WORK

A variety of deep generative models have been proposed for molecular graph generation recently (Segler et al., 2017; Olivecrona et al., 2017; Samanta et al., 2018; Neil et al., 2018). The RVAE model (Ma et al., 2018) used a variational autoencoder for molecule generation, and proposed a novel regularization framework to ensure semantic validity. Jin et al. (2018) proposed to

represent a molecule as a junction tree of chemical scaffolds and proposed the JT-VAE model for molecule generation. For the VAE-based approaches, the optimization of chemical properties is usually done by searching in the latent space with Bayesian Optimization (Jin et al., 2018). De Cao & Kipf (2018) used Generative Adversarial Networks for molecule generation. The state-of-the-art models are built on autoregressive based approaches (You et al., 2018a; Popova et al., 2019). (You et al., 2018a) formulated the problem as a sequential decision process by dynamically adding new nodes and edges based on current sub-graph structures, and the generation policy network is trained by a reinforcement learning framework. Recently, Popova et al. (2019) proposed an autoregressive model called MolecularRNN to generate new nodes and edges based on the generated nodes and edge sequences. The iterative nature of autoregressive model allows effectively leveraging chemical rules for valency checking during generation and hence the proportion of valid molecules generated by these models is very high. However, due to the sequential generation nature, the training process is usually slow. Our GraphAF approach enjoys the advantage of iterative generation process like autoregressive models (the mapping from latent space to observation space) and meanwhile calculates the exact likelihood corresponding to a feedforward neural network (the mapping from observation space to latent space), which can be implemented efficiently through parallel computation.

Two recent work—Graph Normalizing Flows (GNF) (Liu et al., 2019) and GraphNVP (Madhawa et al., 2019)—are also flow-based approaches for graph generation. However, our work is fundamentally different from their work. GNF defines a normalizing flow from a base distribution to the hidden node representations of a pretrained Graph Autoencoders. The generation scheme is done through two separate stages by first generating the node embeddings with the normalizing flow and then generate the graphs based on the generated node embeddings in the first stage. By contrast, in GraphAF, we define an autoregressive flow from a base distribution directly to the molecular graph structures, which can be trained end-to-end. GraphNVP also defines a normalizing flow from a base distribution to the molecular graph structures. However, the generation process of GraphNVP is one-shot, which cannot effectively capture graph structures and also cannot guarantee the validity of generated molecules. In our GraphAF, we formulate the generation process as a sequential decision process and effectively capture the sub-graph structures via graph neural networks, based on which we define a policy function to generate the nodes and edges. The sequential generation process also allows incorporating the chemical rules. As a result, the validity of the generated molecules can be guaranteed. We summarize existing approaches in Table 1.

## 3 PRELIMINARIES

### 3.1 AUTOREGRESSIVE FLOW

A normalizing flow (Kobyzev et al., 2019) defines a parameterized invertible deterministic transformation from a base distribution $\mathcal{E}$ (the latent space, *e.g.*, Gaussian distribution) to real-world observational space $\mathcal{Z}$ (*e.g.* images and speech). Let $f : \mathcal{E} \to \mathcal{Z}$ be an invertible transformation where $\epsilon \sim p_{\mathcal{E}}(\epsilon)$ is the base distribution, then we can compute the density function of real-world data $z$, *i.e.*, $p_Z(z)$, via the change-of-variables formula:

$$p_Z(z) = p_{\mathcal{E}}\left(f_\theta^{-1}(z)\right) \left|\det \frac{\partial f_\theta^{-1}(z)}{\partial z}\right|. \tag{1}$$

Now considering two key processes of normalizing flows as a generative model: (1) Calculating Data Likelihood: given a datapoint $z$, the exact density $p_Z(z)$ can be calculated by inverting the transformation $f$, $\epsilon = f_\theta^{-1}(z)$; (2) Sampling: $z$ can be sampled from the distribution $p_Z(z)$ by first sample $\epsilon \sim p_{\mathcal{E}}(\epsilon)$ and then perform the feedforward transformation $z = f_\theta(\epsilon)$. To efficiently perform the above mentioned operations, $f_\theta$ is required to be invertible with an easily computable Jacobian determinant. Autoregressive flows (AF), originally proposed in Papamakarios et al. (2017), is a variant that satisfies these criteria, which holds a triangular Jacobian matrix, and the determinant can be computed linearly. Formally, given $z \in \mathbb{R}^D$ ($D$ is the dimension of observation data), the autoregressive conditional probabilities can be parameterized as Gaussian distributions:

$$p(z_d|z_{1:d-1}) = \mathcal{N}(z_d|\mu_d, (\alpha_d)^2), \text{ where } \mu_d = g_\mu(z_{1:d-1}; \theta), \alpha_d = g_\alpha(z_{1:d-1}; \theta), \tag{2}$$

where $g_\mu$ and $g_\alpha$ are unconstrained and positive scalar functions of $z_{1:d-1}$ respectively to compute the mean and deviation. In practice, these functions can be implemented as neural networks. The

affine transformation of AF can be written as:

$$f_\theta(\epsilon_d) = z_d = \mu_d + \alpha_d \cdot \epsilon_d; \ f_\theta^{-1}(z_d) = \epsilon_d = \frac{z_d - \mu_d}{\alpha_d}. \tag{3}$$

The Jacobian matrix in AF is triangular, since $\frac{\partial z_i}{\partial \epsilon_j}$ is non-zero only for $j \leq i$. Therefore, the determinant can be efficiently computed through $\prod_{d=1}^D \alpha_d$. Specifically, to perform density estimation, we can apply all individual scalar affine transformations in parallel to compute the base density, each of which depends on previous variables $z_{1:d-1}$; to sample $z$, we can first sample $\epsilon \in \mathbb{R}^D$ and compute $z_1$ through the affine transformation, and then each subsequent $z_d$ can be computed sequentially based on previously observed $z_{1:d-1}$.

## 3.2 GRAPH REPRESENTATION LEARNING

Following existing work, we also represent a molecule as a graph $G = (A, X)$, where $A$ is the adjacency tensor and $X$ is the node feature matrix. Assuming there are $n$ nodes in the graph, $d$ and $b$ are the number of different types of nodes and edges respectively, then $A \in \{0, 1\}^{n \times n \times b}$ and $X \in \{0, 1\}^{n \times d}$. $A_{ijk} = 1$ if there exists a bond with type $k$ between $i^{th}$ and $j^{th}$ nodes.

Graph Convolutional Networks (GCN) (Duvenaud et al., 2015; Gilmer et al., 2017; Kearnes et al., 2016; Kipf & Welling, 2016; Schütt et al., 2017) are a family of neural network architectures for learning representations of graphs. In this paper, we use a variant of Relational GCN (R-GCN) (Schlichtkrull et al., 2018) to learn the node representations (*i.e.*, atoms) of graphs with categorical edge types. Let $k$ denote the embedding dimension. We compute the node embeddings $H^l \in \mathbb{R}^{n \times k}$ at the $l^{th}$ layer of R-GCN by aggregating messages from different edge types:

$$H^l = \text{Agg}\left(\text{ReLU}\left(\{\tilde{D}_i^{-\frac{1}{2}} \tilde{E}_i \tilde{D}_i^{-\frac{1}{2}} H^{l-1} W_i^l\} \big| i \in (1, \ldots, b)\right)\right), \tag{4}$$

where $E_i = A_{[:,:,i]}$ denotes the $i^{th}$ slice of edge-conditioned adjacency tensor, $\tilde{E}_i = E_i + I$, and $\tilde{D}_i = \sum_k \tilde{E}_i[j, k]$. $W_i^{(l)}$ is a trainable weight matrix for the $i^{th}$ edge type. $\text{Agg}(\cdot)$ denotes an aggregation function such as mean pooling or summation. The initial hidden node representation $H^0$ is set as the original node feature matrix $X$. After $L$ message passing layers, we use the the final hidden representation $H^L$ as the node representations. Meanwhile, the whole graph representations can be defined by aggregating the whole node representations using a readout function (Hamilton et al., 2017), *e.g.*, summation.

# 4 PROPOSED METHOD

## 4.1 GRAPHAF FRAMEWORK

Similar to existing works like GCPN (You et al., 2018a) and MolecularRNN (Popova et al., 2019), we formalize the problem of molecular graph generation as a sequential decision process. Let $G = (A, X)$ denote a molecular graph structure. Starting from an empty graph $G_1$, in each step a new node $X_i$ is generated based on the current sub-graph structure $G_i$, *i.e.*, $p(X_i | G_i)$. Afterwards, the edges between this new node and existing nodes are sequentially generated according to the current graph structure, *i.e.*, $p(A_{ij} | G_i, X_i, A_{i,1:j-1})$. This process is repeated until all the nodes and edges are generated. An illustrative example is given in Fig. 1(a).

GraphAF is aimed at defining an invertible transformation from a base distribution (e.g. multivariate Gaussian) to a molecular graph structure $G = (A, X)$. Note that we add one additional type of edge between two nodes, which corresponds to *no edge* between two nodes, *i.e.*, $A \in \{0, 1\}^{n \times n \times (b+1)}$. Since both the node type $X_i$ and the edge type $A_{ij}$ are discrete, which do not fit into a flow-based model, a standard approach is to use *Dequantization* technique (Dinh et al., 2016; Kingma & Dhariwal, 2018) to convert discrete data into continuous data by adding real-valued noise. We follow this approach to preprocess a discrete graph $G = (A, X)$ into continuous data $z = (z^A, z^X)$:

$$z_i^X = X_i + u, \ u \sim U[0, 1)^d; \ z_{ij}^A = A_{ij} + u, \ u \sim U[0, 1)^{b+1}. \tag{5}$$

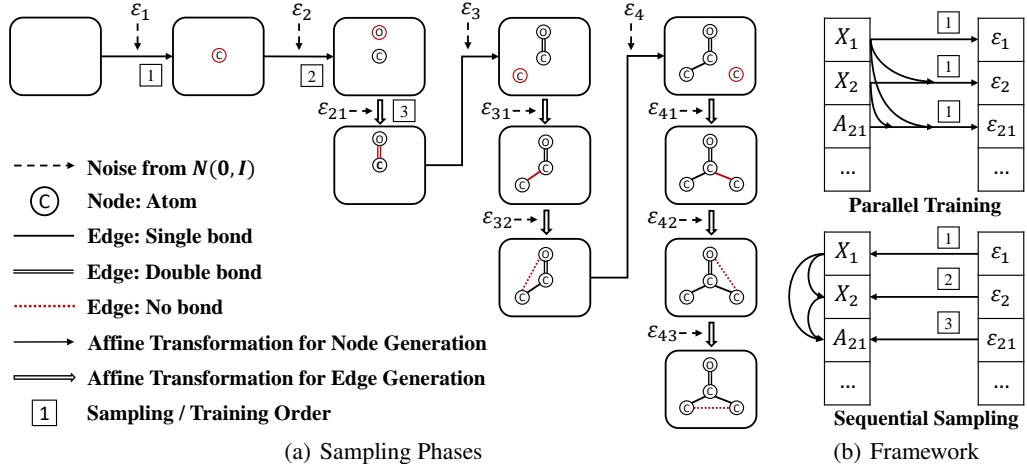

Figure 1: Overview of the proposed GraphAF model. (a) Illustration of the generative procedure. New nodes or edges are marked in red. Starting from an empty graph and iteratively sample random variables to map them to atom/bond features. The numbered first three steps correspond to the maps in the bottom figure of Fig. 1(b). (b) Computation graph of GraphAF. The left side are the nodes and edges and the right are latent variables.

We present further discussions on dequantization techniques in Appendix A. Formally, we define the conditional distributions for the generation as:

$$p(z_i^X|G_i) = \mathcal{N}(\mu_i^X, (\alpha_i^X)^2), \tag{6}$$
$$\text{where } \mu_i^X = g_{\mu^X}(G_i), \alpha_i^X = g_{\alpha^X}(G_i),$$
$$p(z_{ij}^A|G_i, X_i, A_{i,1:j-1}) = \mathcal{N}(\mu_{ij}^A, (\alpha_{ij}^A)^2), \ j \in \{1, 2, \ldots, i-1\}, \tag{7}$$
$$\text{where } \mu_{ij}^A = g_{\mu^A}(G_i, X_i, A_{i,1:j-1}), \alpha_{ij}^A = g_{\alpha^A}(G_i, X_i, A_{i,1:j-1}),$$

where $g_{\mu^X}$, $g_{\mu^A}$ and $g_{\alpha^X}$, $g_{\alpha^A}$ are parameterized neural networks for defining the mean and standard deviation of a Gaussian distribution. More specifically, given the current sub-graph structure $G_i$, we use a $L$-layer of Relational GCN (defined in Section 3.2) to learn the node embeddings $H_i^L \in \mathbb{R}^{n \times k}$, and the embedding of entire sub-graph $\tilde{h}_i \in \mathbb{R}^k$, based on which we define the mean and standard deviations of Gaussian distributions to generate the nodes and edges respectively:

$$\text{R-GCN: } H_i^L = \text{R-GCN}(G_i), \ \tilde{h}_i = \text{sum}(H_i^L);$$
$$\text{Node-MLPs: } g_{\mu^X} = m_{\mu^X}(\tilde{h}_i), \ g_{\alpha^X} = m_{\alpha^X}(\tilde{h}_i); \tag{8}$$
$$\text{Edge-MLPs: } g_{\mu^A} = m_{\mu^A}(\tilde{h}_i, H_{i,i}^L, H_{i,j}^L), \ g_{\alpha^A} = m_{\alpha^A}(\tilde{h}_i, H_{i,i}^L, H_{i,j}^L),$$

where sum denotes the sum-pooling operation, and $H_{i,j}^L \in \mathbb{R}^k$ denotes the embedding of the $j$-th node in the embeddings $H_i^L$. $m_{\mu^X}, m_{\alpha^X}$ are Multi-Layer Perceptrons (MLP) that predict the node types according to the current sub-graph embedding. and $m_{\mu^A}, m_{\alpha^A}$ are MLPs that predict the types of edges according to the current sub-graph embedding and node embeddings.

To generate a new node $X_i$ and its edges connected to existing nodes, we just sample random variables $\epsilon_i$ and $\epsilon_{ij}$ from the base Gaussian distribution and convert it to discrete features. More specifically,

$$z_i^X = \epsilon_i \odot \alpha_i^X + \mu_i^X, \ \epsilon_i \in \mathbb{R}^d;$$
$$z_{ij}^A = \epsilon_{ij} \odot \alpha_{ij}^A + \mu_{ij}^A, \ j \in \{1, 2, \ldots, i-1\}, \ \epsilon_{ij} \in \mathbb{R}^{b+1}, \tag{9}$$

where $\odot$ is the element-wise multiplication. In practice, a real molecular graph is generated by taking the argmax of generated continuous vectors, *i.e.*, $X_i = v_{\text{argmax}(z_i^X)}^d$ and $A_{ij} = v_{\text{argmax}(z_{ij}^A)}^{b+1}$, where $v_q^p$ denotes a $p$ dimensional one-hot vector with $q^{\text{th}}$ dimension equal to 1.

Let $\epsilon = \{\epsilon_1, \epsilon_2, \epsilon_{21}, \epsilon_3, \epsilon_{31}, \epsilon_{32}, \ldots, \epsilon_n, \epsilon_{n1}, \ldots, \epsilon_{n,n-1}\}$, where $n$ is the number of atoms in the given molecule, GraphAF defines an invertible mapping between the base Gaussian distribution $\epsilon$

and the molecule structures $z = (z^A, z^X)$. According to Eq. 9, the inverse process from $z = (z^A, z^X)$ to $\epsilon$ can be easily calculated as:

$$\epsilon_i = \left(z_i^X - \mu_i^X\right) \odot \frac{1}{\alpha_i^X}; \quad \epsilon_{ij} = \left(z_{ij}^A - \mu_{ij}^A\right) \odot \frac{1}{\alpha_{ij}^A}, j \in \{1, 2, \ldots, i-1\}, \tag{10}$$

where $\frac{1}{\alpha_i^X}$ and $\frac{1}{\alpha_{ij}^A}$ denote element-wise reciprocals of $\alpha_i^X$ and $\alpha_{ij}^A$ respectively.

## 4.2 EFFICIENT PARALLEL TRAINING

In GraphAF, since $f : \mathcal{E} \to \mathcal{Z}$ is autoregressive, the Jacobian matrix of the inverse process $f^{-1} : \mathcal{Z} \to \mathcal{E}$ is a triangular matrix, and its determinant can be calculated very efficiently. Given a mini-batch of training data $G$, the exact density of each molecule under a given order can be efficiently computed by the change-of-variables formula in Eq. 1. Our objective is to maximize the likelihood of training data.

During training, we are able to perform parallel computation by defining a feedforward neural network between the input molecule graph $G$ and the output latent variable $\epsilon$ by using *masking*. The mask drops out some connections from inputs to ensure that R-GCN is only connected to the sub-graph $G_i$ when inferring the hidden variable of node $i$, *i.e.*, $\epsilon_i$, and connected to sub-graph $G_i, X_i, A_{i,1:j-1}$ when inferring the hidden variable of edge $A_{ij}$, *i.e.*, $\epsilon_{ij}$. This is similar to the approaches used in MADE (Germain et al., 2015) and MAF (Papamakarios et al., 2017). With the masking technique, GraphAF satisfies the autoregressive property, and at the same time $p(G)$ can be efficiently calculated in just one forward pass by computing all the conditionals in parallel.

To further accelerate the training process, the nodes and edges of a training graph are re-ordered according to the breadth-first search (BFS) order, which is widely adopted by existing approaches for graph generation (You et al., 2018b; Popova et al., 2019). Due to the nature of BFS, bonds can only be present between nodes within the same or consecutive BFS depths. Therefore, the maximum dependency distance between nodes is bounded by the largest number of nodes in a single BFS depth. In our data sets, any single BFS depth contains no more than 12 nodes, which means we only need to model the edges between current atom and the latest generated 12 atoms.

Due to space limitation, we summarize the detailed training algorithm into Appendix B.

## 4.3 VALIDITY CONSTRAINED SAMPLING

In chemistry, there exist many chemical rules, which can help to generate valid molecules. Thanks to the sequential generation process, GraphAF can leverage these rules in each generation step. Specifically, we can explicitly apply a valency constraint during sampling to check whether current bonds have exceeded the allowed valency, which has been widely adopted in previous models (You et al., 2018a; Popova et al., 2019). Let $|A_{ij}|$ denote the order of the chemical bond $A_{ij}$. In each edge generation step of $A_{ij}$, we check the following valency constraint for the $i^{th}$ and $j^{th}$ atoms:

$$\sum_j |A_{ij}| \leq \text{Valency}(X_i) \text{ and } \sum_i |A_{ij}| \leq \text{Valency}(X_j). \tag{11}$$

If the newly added bond breaks the valency constraint, we just reject the bond $A_{ij}$, sample a new $\epsilon_{ij}$ in the latent space and generate another new bond type. The generation process will terminate if one of the following conditions is satisfied: 1) the graph size reaches the max-size $n$, 2) no bond is generated between the newly generated atom and previous sub-graph. Finally, hydrogens are added to the atoms that have not filled up their valencies.

## 4.4 GOAL-DIRECTED MOLECULE GENERATION WITH REINFORCEMENT LEARNING

So far, we have introduced how to use GraphAF to model the data density of molecular graph structures and generate valid molecules. Nonetheless, for drug discovery, we also need to optimize the chemical properties of generated molecules. In this part, we introduce how to fine-tune our generation process with reinforcement learning to optimize the properties of generated molecules.

**State and Policy Network.** The state is the current sub-graph, and the initial state is an empty graph. The policy network is the same as the autoregressive model defined in Section 4.1, which

includes the process of generating a new atom based on the current sub-graph and generating the edges between the new atom and existing atoms, *i.e.*, $p(X_i|G_i)$ and $p(A_{ij}|G_i, X_i, A_{i,1:j-1})$. The policy network itself defines a distribution $p_\theta$ of molecular graphs $G$. If there are no edges between the newly generated atom and current sub-graph, the generation process terminates. For the state transition dynamics, we also incorporate the valency check constraint.

**Reward design.** Similar to GCPN You et al. (2018a), we also incorporate both intermediate and final rewards for training the policy network. A small penalization will be introduced as the intermediate reward if the edge predictions violate the valency check. The final rewards include both the score of targeted-properties of generated molecules such as octanol-water partition coefficient (logP) or drug-likeness (QED) (Bickerton et al., 2012) and the chemical validity reward such as penalties for molecules with excessive steric strain and or functional groups that violate ZINC functional group filters (Irwin et al., 2012). The final reward is distributed to all intermediate steps with a discounting factor to stabilize the training.

In practice, we adopt Proximal Policy Optimization (PPO) (Schulman et al., 2017), an advanced policy gradient algorithm to train GraphAF in the above defined environment. Let $G_{ij}$ be the shorthand notation of sub-graph $G_i \cup X_i \cup A_{i,1:j-1}$. Formally,

$$
\begin{aligned}
L(\theta) = - E_{G \sim p_\theta} \Big\{ E_i \Big[ &\min\left(r_i(\theta)V(G_i, X_i), \text{clip}\left(r_i(\theta), 1-\epsilon, 1+\epsilon\right)V(G_i, X_i)\right) \\
&+ E_j\big[\min\left(r_{ij}(\theta)V(G_{ij}, A_{ij}), \text{clip}\left(r_{ij}(\theta), 1-\epsilon, 1+\epsilon\right)V(G_{ij}, A_{ij})\right)\big]\Big]\Big\},
\end{aligned} \tag{12}
$$

where $r_i(\theta) = \frac{p_\theta(X_i|G_i)}{p_{\theta_{old}}(X_i|G_i)}$ and $r_{ij}(\theta) = \frac{p_\theta(A_{ij}|G_{ij})}{p_{\theta_{old}}(A_{ij}|G_{ij})}$ are ratios of probabilities output by old and new policies, and $V(state, action)$ is the estimated advantage function with a moving average baseline to reduce the variance. More specifically, we treat generating a node and all its edges with existing nodes as one step and maintain a moving average baseline for each step. The clipped surrogate objective prevents the policy from being updated to collapse for some extreme rewards.

## 5 EXPERIMENTS

### 5.1 EXPERIMENT SETUP

**Evaluation Tasks.** Following existing works on molecule generation (Jin et al., 2018; You et al., 2018a; Popova et al., 2019), we conduct experiments by comparing with the state-of-the-art approaches on three standard tasks. **Density Modeling and Generation** evaluates the model's capacity to learn the data distribution and generate realistic and diverse molecules. **Property Optimization** concentrates on generating novel molecules with optimized chemical properties. For this task, we fine-tune our network pretrained from the density modeling task to maximize the desired properties. **Constrained Property Optimization** is first proposed in Jin et al. (2018), which is aimed at modifying the given molecule to improve desired properties while satisfying a similarity constraint.

**Data.** We use the ZINC250k molecular dataset (Irwin et al., 2012) for training. The dataset contains $250,000$ drug-like molecules with a maximum atom number of 38. It has 9 atom types and 3 edge types. We use the open-source chemical software RDkit (Landrum, 2016) to preprocess molecules. All molecules are presented in kekulized form with hydrogen removed.

**Baselines.** We compare GraphAF with the following state-of-the-art approaches for molecule generation. **JT-VAE** (Jin et al., 2018) is a VAE-based model which generates molecules by first decoding a tree structure of scaffolds and then assembling them into molecules. JT-VAE has been shown to outperform other previous VAE-based models (Kusner et al., 2017; Gómez-Bombarelli et al., 2018; Simonovsky & Komodakis, 2018). **GCPN** is a state-of-the-art approach which combines reinforcement learning and graph representation learning methods to explore the vast chemical space. **MolecularRNN (MRNN)**, another autoregressive model, uses RNN to generate molecules in a sequential manner. We also compare our model with **GraphNVP** (Madhawa et al., 2019), a recently proposed flow-based model. Results of baselines are taken from original papers unless stated.

**Implementation Details.** GraphAF is implemented in PyTorch (Paszke et al., 2017). The R-GCN is implemented with 3 layers, and the embedding dimension is set as 128. The max graph size is set as 48 empirically. For density modeling, we train our model for 10 epochs with a batch size of 32 and a

Table 2: Comparison of different models on density modeling and generation. *Reconstruction* is only evaluated on latent variable models. *Validity w/o check* is only evaluated on models with valency constraints. Result with † is obtained by running GCPN's open-source code. Results with ‡ are taken from Popova et al. (2019).

| Method | Validity | Validity w/o check | Uniqueness | Novelty | Reconstruction |
|--------|----------|--------------------|------------|---------|----------------|
| JT-VAE | 100% | — | 100%[‡] | 100%[‡] | 76.7% |
| GCPN | 100% | 20%[†] | 99.97%[‡] | 100%[‡] | — |
| MRNN | 100% | 65% | 99.89% | 100% | — |
| GraphNVP | 42.60% | — | 94.80% | 100% | 100% |
| GraphAF | 100% | 68% | 99.10% | 100% | 100% |

Table 3: Results of density modeling and generation on three different datasets.

| Method | Validity | Validity w/o check | Uniqueness | Novelty | Reconstruction |
|--------|----------|--------------------|------------|---------|----------------|
| ZINC250k | 100% | 68% | 99.10% | 100% | 100% |
| QM9 | 100% | 67% | 94.51% | 88.83% | 100% |
| MOSES | 100% | 71% | 99.99% | 100% | 100% |

learning rate of 0.001. For property optimization, we perform a grid search on the hyperparameters and select the best setting according to the chemical scoring performance. We use Adam (Kingma & Ba, 2014) to optimize our model. Full training details can be found in Appendix C.

## 5.2 NUMERICAL RESULTS

**Density Modeling and Generation.** We evaluate the ability of the proposed method to model real molecules by utilizing the widely-used metrics: *Validity* is the percentage of valid molecules among all the generated graphs. *Uniqueness* is the percentage of unique molecules among all the generated molecules. *Novelty* is the percentage of generated molecules not appearing in training set. *Reconstruction* is the percentage of the molecules that can be reconstructed from latent vectors. We calculate the above metrics from 10,000 randomly generated molecules.

Table 2 shows that GraphAF achieves competitive results on all four metrics. As a flow-based model, GraphAF holds perfect reconstruction ability compared with VAE approaches. Our model also achieves a 100% validity rate since we can leverage the valency check during sequential generation. By contrast, the validity rate of another flow-based approach GraphNVP is only 42.60% due to its one-shot sampling process. An interesting result is that even without the valency check during generation, GraphAF can still achieve a validity rate as high as 68%, while previous state-of-the-art approach GCPN only achieves 20%. This indicates the strong flexibility of GraphAF to model the data density and capture the domain knowledge from unsupervised training on the large chemical dataset. We also compare the efficiency of different methods on the same computation environment, a machine with 1 Tesla V100 GPU and 32 CPU cores. To achieve the results in Table 2, JT-VAE and GCPN take around 24 and 8 hours, respectively, while GraphAF only takes 4 hours.

To show that GraphAF is not overfitted to the specific dataset ZINC250k, we also conduct experiments on two other molecule datasets, QM9 (Ramakrishnan et al., 2014) and MOSES (Polykovskiy et al., 2018). QM9 contains 134k molecules with 9 heavy atoms, and MOSES is much larger and more challenging, which contains 1.9M molecules with up to 30 heavy atoms. Table 3 shows that GraphAF can always generate valid and novel molecules even on the more complicated dataset.

Furthermore, though GraphAF is originally designed for molecular graph generation, it is actually very general and can be used to model different types of graphs by simply modifying the node and edge generating functions Edge-MLPs and Node-MLPs (Eq. 8). Following the experimental setup of Graph Normalizing Flows (GNF) (Liu et al., 2019), we test GraphAF on two generic graph datasets: COMMUNITY-SMALL, which is a synthetic data set containing 100 2-community graphs, and EGO-SMALL, which is a set of graphs extracted from Citeseer dataset (Sen et al., 2008). In practice, we use one-hot indicator vectors as node features for R-GCN. We borrow open source scripts from GraphRNN (You et al., 2018b) to generate datasets and evaluate different models. For evaluation, we report the Maximum Mean Discrepancy (MMD) (Gretton et al., 2012) between generated

Table 4: Comparison between different graph generative models on general graphs with MMD metrics. We follow the evaluation scheme of GNF (Liu et al., 2019). Results of baselines are also taken from GNF.

| Method | COMMUNITY-SMALL | | | EGO-SMALL | | |
|---|---|---|---|---|---|---|
| | Degree | Cluster | Orbit | Degree | Cluster | Orbit |
| GraphVAE | 0.35 | 0.98 | 0.54 | 0.13 | 0.17 | 0.05 |
| DEEPGMG | 0.22 | 0.95 | 0.4 | 0.04 | 0.10 | 0.02 |
| GraphRNN | 0.08 | 0.12 | 0.04 | 0.09 | 0.22 | 0.003 |
| GNF | 0.20 | 0.20 | 0.11 | 0.03 | 0.10 | 0.001 |
| GraphAF | 0.18 | 0.20 | 0.02 | 0.03 | 0.11 | 0.001 |
| GraphRNN(1024) | 0.03 | 0.01 | 0.01 | 0.04 | 0.05 | 0.06 |
| GNF(1024) | 0.12 | 0.15 | 0.02 | 0.01 | 0.03 | 0.0008 |
| GraphAF(1024) | 0.06 | 0.10 | 0.015 | 0.04 | 0.04 | 0.008 |

Table 5: Comparison of the top 3 property scores of generated molecules.

| Method | Penalized logP | | | | QED | | | |
|---|---|---|---|---|---|---|---|---|
| | 1st | 2nd | 3rd | Validity | 1st | 2nd | 3rd | Validity |
| ZINC (Dataset) | 4.52 | 4.30 | 4.23 | 100.0% | 0.948 | 0.948 | 0.948 | 100.0% |
| JT-VAE (Jin et al., 2018) | 5.30 | 4.93 | 4.49 | 100.0% | 0.925 | 0.911 | 0.910 | 100.0% |
| GCPN (You et al., 2018a) | 7.98 | 7.85 | 7.80 | 100.0% | **0.948** | 0.947 | 0.946 | 100.0% |
| MRNN[1] (Popova et al., 2019) | 8.63 | 6.08 | 4.73 | 100.0% | 0.844 | 0.796 | 0.736 | 100.0% |
| GraphAF | **12.23** | **11.29** | **11.05** | 100.0% | **0.948** | **0.948** | **0.947** | 100.0% |

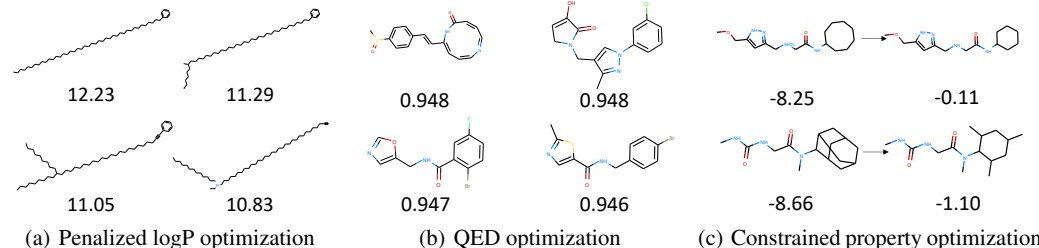

| (a) Penalized logP optimization | (b) QED optimization | (c) Constrained property optimization |
|---|---|---|

Figure 2: Molecules generated in property optimization and constrained property optimization tasks. (a) Molecules with high penalized logP scores. (b) Molecules with high QED scores. (c) Two pairs of molecules in constrained property optimization for penalized logP with similarity 0.71(top) and 0.64(bottom).

and training graphs using some specific metrics on graphs proposed by You et al. (2018b). The results in Table 4 demonstrate that when applied to generic graphs, GraphAF can still consistently yield comparable or better results compared with GraphRNN and GNF. We give the visualization of generated generic graphs in Appendix D.

**Property Optimization.** In this task, we aim at generating molecules with desired properties. Specifically, we choose penalized logP and QED as our target property. The former score is logP score penalized by ring size and synthetic accessibility, while the latter one measures the drug-likeness of the molecules. Note that both scores are calculated using empirical prediction models and we adopt the script used in (You et al., 2018a) to make results comparable. To perform this task, we pretrain the GraphAF network for 300 epochs for likelihood modeling, and then apply the RL process described in section 4.4 to fine-tune the network towards desired chemical properties. Detailed reward design and hyper-parameters setting can be found in Appendix C. Following existing works, we report the top-3 scores found by each model.

---

[1]The scores reported here are recalculated based on top 3 molecules presented in the original paper (Popova et al., 2019) using GCPN's script.

Table 6: Comparison of results on constrained property optimization.

| $\delta$ | JT-VAE | | | GCPN | | | GraphAF | | |
|---|---|---|---|---|---|---|---|---|---|
| | Improvement | Similarity | Success | Improvement | Similarity | Success | Improvement | Similarity | Success |
| 0.0 | $1.91 \pm 2.04$ | $0.28 \pm 0.15$ | 97.5% | $4.20 \pm 1.28$ | $0.32 \pm 0.12$ | 100% | $\mathbf{13.13 \pm 6.89}$ | $0.29 \pm 0.15$ | 100% |
| 0.2 | $1.68 \pm 1.85$ | $0.33 \pm 0.13$ | 97.1% | $4.12 \pm 1.19$ | $0.34 \pm 0.11$ | 100% | $\mathbf{11.90 \pm 6.86}$ | $0.33 \pm 0.12$ | 100% |
| 0.4 | $0.84 \pm 1.45$ | $0.51 \pm 0.10$ | 83.6% | $2.49 \pm 1.30$ | $0.47 \pm 0.08$ | 100% | $\mathbf{8.21 \pm 6.51}$ | $0.49 \pm 0.09$ | 99.88% |
| 0.6 | $0.21 \pm 0.71$ | $0.69 \pm 0.06$ | 46.4% | $0.79 \pm 0.63$ | $0.68 \pm 0.08$ | 100% | $\mathbf{4.98 \pm 6.49}$ | $0.66 \pm 0.05$ | 96.88% |

As shown in Table 5, GraphAF outperforms all baselines by a large margin for penalized logP score and achieves comparable results for QED. This phenomenon indicates that combined with RL process, GraphAF successfully captures the distribution of desired molecules. Note that we re-evaluate the properties of the top-3 molecules found by MolecularRNN, which turn out to be lower than the results reported in the original paper. Figure 2(a) and 2(b) show the molecules with the highest score discovered by our model. More realistic molecules generated by GraphAF with penalized logP score ranging from 5 to 10 are presented in Figure 6 in Appendix E.

One should note that, as defined in Sec 4.4, our RL process is close to the one used in previous work GCPN (You et al., 2018a). Therefore, the good property optimization performance is believed to come from the flexibility of flow. Compared with the GAN model used in GCPN, which is known to suffer from the mode collapse problem, flow is flexible at modeling complex distribution and generating diverse data (as shown in Table 2 and Table 3). This allows GraphAF to explore a variety of molecule structures in the RL process for molecule properties optimization.

**Constrained Property Optimization.** The goal of the last task is to modify the given molecule to improve specified property with the constraint that the similarity between the original and modified molecule is above a threshold $\delta$. Following Jin et al. (2018) and You et al. (2018a), we choose to optimize penalized logP for 800 molecules in ZINC250k with the *lowest* scores and adopt Tanimoto similarity with Morgan fingerprint (Rogers & Hahn, 2010) as the similarity metric.

Similar to the property optimization task, we pretrain GraphAF via density modeling and then fine-tune the model with RL. During generation, we set the initial states as sub-graphs randomly sampled from 800 molecules to be optimized. For evaluation, we report the mean and standard deviation of the highest improvement and the corresponding similarity between the original and modified molecules in Table 6. Experiment results show that GraphAF significantly outperforms all previous approaches and almost always succeeds in improving the target property. Figure 2(c) visualizes two optimization examples, showing that our model is able to improve the penalized logP score by a large margin while maintaining a high similarity between the original and modified molecule.

## 6 CONCLUSION

We proposed GraphAF, the first flow-based autoregressive model for generating realistic and diverse molecular graphs. GraphAF is capable to model the complex molecular distribution thanks to the flexibility of normalizing flow, as well as generate novel and 100% valid molecules in empirical experiments. Moreover, the training of GraphAF is very efficient. To optimize the properties of generated molecules, we fine-tuned the generative process with reinforcement learning. Experimental results show that GraphAF outperforms all previous state-of-the-art baselines on the standard tasks. In the future, we plan to train our GraphAF model on larger datasets and also extend it to generate other types of graph structures (*e.g.*, social networks).

ACKNOWLEDGEMENT

We would like to thank Min Lin, Meng Qu, Andreea Deac, Laurent Dinh, Louis-Pascal A. C. Xhonneux and Vikas Verma for the extremely helpful discussions and comments. This project is supported by the Natural Sciences and Engineering Research Council of Canada, the Canada CIFAR AI Chair Program, and a collaboration grant between Microsoft Research and Mila. Ming Zhang is supported by National Key Research and Development Program of China with Grant No. 2018AAA0101900/2018AAA0101902 as well as Beijing Municipal Commission of Science and Technology under Grant No. Z181100008918005. Weinan Zhang is supported by National Natural Science Foundation of China (61702327, 61772333, 61632017, 81771937) and Shanghai Sailing Program (17YF1428200).

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

# Appendix

## A    DISCCUSIONS ON DEQUANTIZATION TECHNIQUES

The dequantization techniques allow mapping the discrete data into the continuous one by adding a small noise to each dimension. By adding noise from $U[0, 1)$, we can ensure that the range of different categories will not overlap. For example, after dequantization, the value of 1-entry in the one-hot vector lies in $[1, 2)$ while the 0-entry lies in $[0, 1)$. Therefore, we can map the dequantized continuous data back to the discrete one-hot data by easily performing the argmax operation in the generation process. Theoretically, as shown in Theis et al. (2016); Ho et al. (2019), training a continuous density model on uniform dequantized data can be interpreted as maximizing a lower bound on the log-likelihood for the original discrete data. Mathematically, this statement holds for both image data and binary/categorical data.

Furthermore, as suggested in Ho et al. (2019), instead of adding random uniform noise to each discrete data for dequantization, a more advanced dequantization technique is to treat the noise as a hidden variable and use variational inference to infer the optimum noise added to each discrete data, which we would like to explore in our future work.

## B    PARALLEL TRAINING ALGORITHM

---

**Algorithm 1** Parallel Training Algorithm of GraphAF

---

**Input**: $\eta$ learning rate, $M$ batch size, $P$ maximum dependency distance in BFS, Adam hyperparameters $\beta_1, \beta_2$, use $\text{Prod}(\cdot)$ as the product of elements across dimensions of a tensor
**Initial**: Parameters $\theta$ of GraphAF (R-GCN, Node-MLP and Edge-MLP)

1: **while** $\theta$ is not converged **do**
2:    **for** $m = 1, ..., M$ **do**
3:       Sample a molecule $mol$ from dataset and get the graph size $N$
4:       Convert $mol$ to $G = (A, X)$ with BFS re-ordering
5:       **for** $i = 1, ..., N$ **do**
6:          $z_i^X = X_i + u,\ u \sim U[0, 1)^d$
7:          $\mu_i^X = g_{\mu^X}(G_i),\ \alpha_i^X = g_{\alpha^X}(G_i)$
8:          $\epsilon_i = (z_i^X - \mu_i^X) \odot \frac{1}{\alpha_i^X}$
9:          $\mathcal{L}_i^X = -\log(\text{Prod}(p_{\mathcal{E}}(\epsilon_i))) - \log(\text{Prod}(\frac{1}{\alpha_i^X}))$
10:          **for** $j = \max\{1, i - P\}, ..., i - 1$ **do**
11:             $z_{ij}^A = A_{ij} + u,\ u \sim U[0, 1)^{b+1}$
12:             $\mu_{ij}^A = g_{\mu^A}(G_i, X_i, A_{i,1:j-1}),\ \alpha_{ij}^A = g_{\alpha^A}(G_i, X_i, A_{i,1:j-1})$
13:             $\epsilon_{ij} = (z_{ij}^A - \mu_{ij}^A) \odot \frac{1}{\alpha_{ij}^A}$
14:             $\mathcal{L}_{ij}^A = -\log(\text{Prod}(p_{\mathcal{E}}(\epsilon_{ij}))) - \log(\text{Prod}(\frac{1}{\alpha_{ij}^A}))$
15:          **end for**
16:       **end for**
17:       $\mathcal{L}_m^G = \sum_{i=1}^n \left(\mathcal{L}_i^X + \sum_{j=1}^P \mathcal{L}_{ij}^A\right)$
18:    **end for**
19:    $\theta \leftarrow \text{ADAM}(\frac{1}{M}\sum_{m=1}^m \mathcal{L}_m^G, \theta, \eta, \beta_1, \beta_2)$
20: **end while**

---

## C    EXPERIMENT DETAILS

**Network architecture.**    The network architecture is fixed among all three tasks. More specifically, the R-GCN is implemented with 3 layers and the embedding dimension is set as 128. We use batch normalization before graph pooling to accelerate the convergence and choose sum-pooling as

the readout function for graph representations. Both node MLPs and edge MLPs have two fully-connected layers equipped with $\tanh$ non-linearity.

**Density Modeling and Generation.** To achieve the results in Table 2, we train GraphAF on ZINC250K with a batch size of 32 on 1 Tesla V100 GPU and 32 CPU cores for 10 epochs. We optimize our model with Adam with a fixed learning rate of 0.001.

**Property Optimization.** For both property optimization and constrained property optimization, we first pretrain a GraphAF network via the density modeling task for 300 epochs, and then fine-tune the network toward desired molecular distribution through RL process. Following are details about the reward design for property optimization. The reward of each step consists of step-wise validity rewards and the final rewards discounted by a fixed factor $\gamma$. The step-wise validity penalty is fixed as -1. The final reward of a molecule $m$ includes both property-targeted reward and chemical validation reward. We adopt the same chemical validation rewards as GCPN. We define property-targeted reward as follows:

$$r(m) = t_1 \cdot QED(m)$$
$$r(m) = \exp\left(\frac{logP_{pen}(mol)}{t_2}\right) \tag{13}$$

$\gamma$ is set to 0.97 for QED optimization and 0.9 for penalized logP optimization respectively. We fine-tune the pretrained model for 200 iterations with a fixed batch size of 64 using Adam optimizer. We also adopt a linear learning rate warm-up to stabilize the training. We perform the grid search to determine the optimal hyperparameters according to the chemical scoring performance. The search space is summarised in Table 7.

Table 7: Tuned-parameters for policy gradient and their search space.

| PARAM | Description | Search space |
|---|---|---|
| $lr$ | Learning rate | $\{0.001, 0.0005, 0.0001\}$ |
| $t_1$ | Coefficient for QED score | $\{2, 3, 4, 5\}$ |
| $t_2$ | Temperature for exponential function | $\{3, 4, 5\}$ |
| $wm$ | Number of warm up iterations | $\{12, 24, 36\}$ |

**Constrained Property Optimization.** We first introduce the way we sample sub-graphs from 800 ZINC molecules. Given a molecule, we first randomly sample a BFS order and then drop the last $m$ nodes in BFS order as well as edges induced by these nodes, where $m$ is randomly chosen from $\{0, 1, 2, 3, 4, 5\}$ each time. Finally, we reconstruct the sub-graph from the remaining nodes in the BFS sequence. Note that the final sub-graph is connected due to the nature of BFS order. For reward design, we set it as the improvement of the target score. We fine-tune the pretrained model for 200 iterations with a batch size of 64. We also use Adam with a learning rate of 0.0001 to optimize the model. Finally, each molecule is optimized for 200 times by the tuned model.

## D    VISUALIZATION OF GENERATED GENERIC GRAPHS

We present visualizations of graphs from both the training set and generated graphs by GraphAF in Figure 3 and Figure 4. The visualizations demonstrate that GraphAF has strong ability to model different graph structures in the generic graph datasets.

## E    MORE MOLECULE SAMPLES

We present more molecule samples generated by GraphAF in the following pages. Figure 5 presents 50 molecules randomly sampled from multivariate Gaussian, which justify the ability of our model to generate novel, realistic and unique molecules. From Figure 6 we can see that our model is able to generate molecules with high and diverse penalized logP scores ranging from 5 to 10. For constrained property optimization of penalized logP score, as shown by Figure 7, our model can either reduce the ring size, remove the big ring or grow carbon chains from the original molecule, improving the penalized logP score by a large margin.

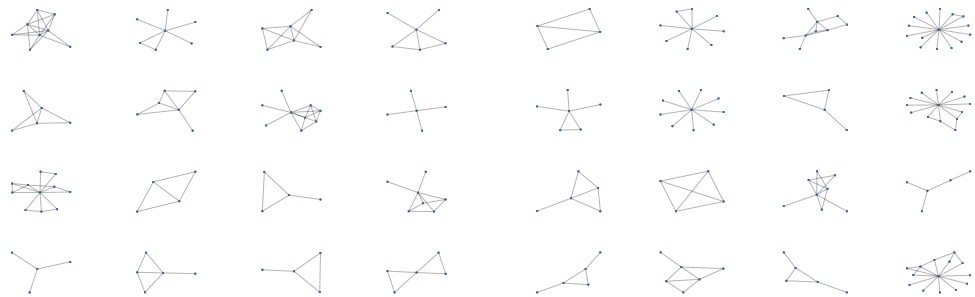

(a) Graphs from training set                    (b) Graphs generated by GraphAF

Figure 3: Visualizations of training graphs and generated graphs of EGO-SMALL.

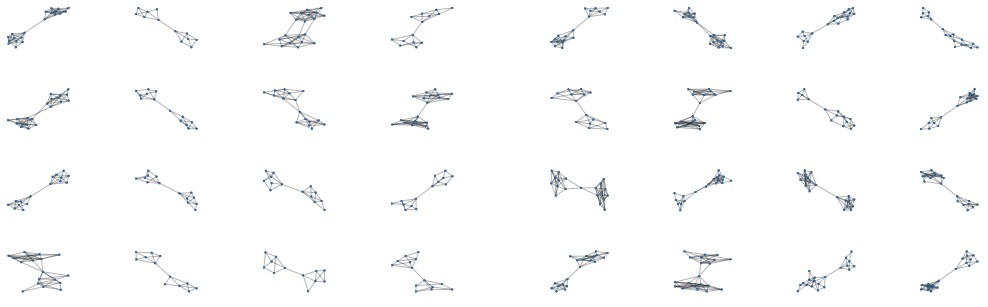

(a) Graphs from training set                    (b) Graphs generated by GraphAF

Figure 4: Visualizations of training graphs and generated graphs of COMMUNITY-SMALL.

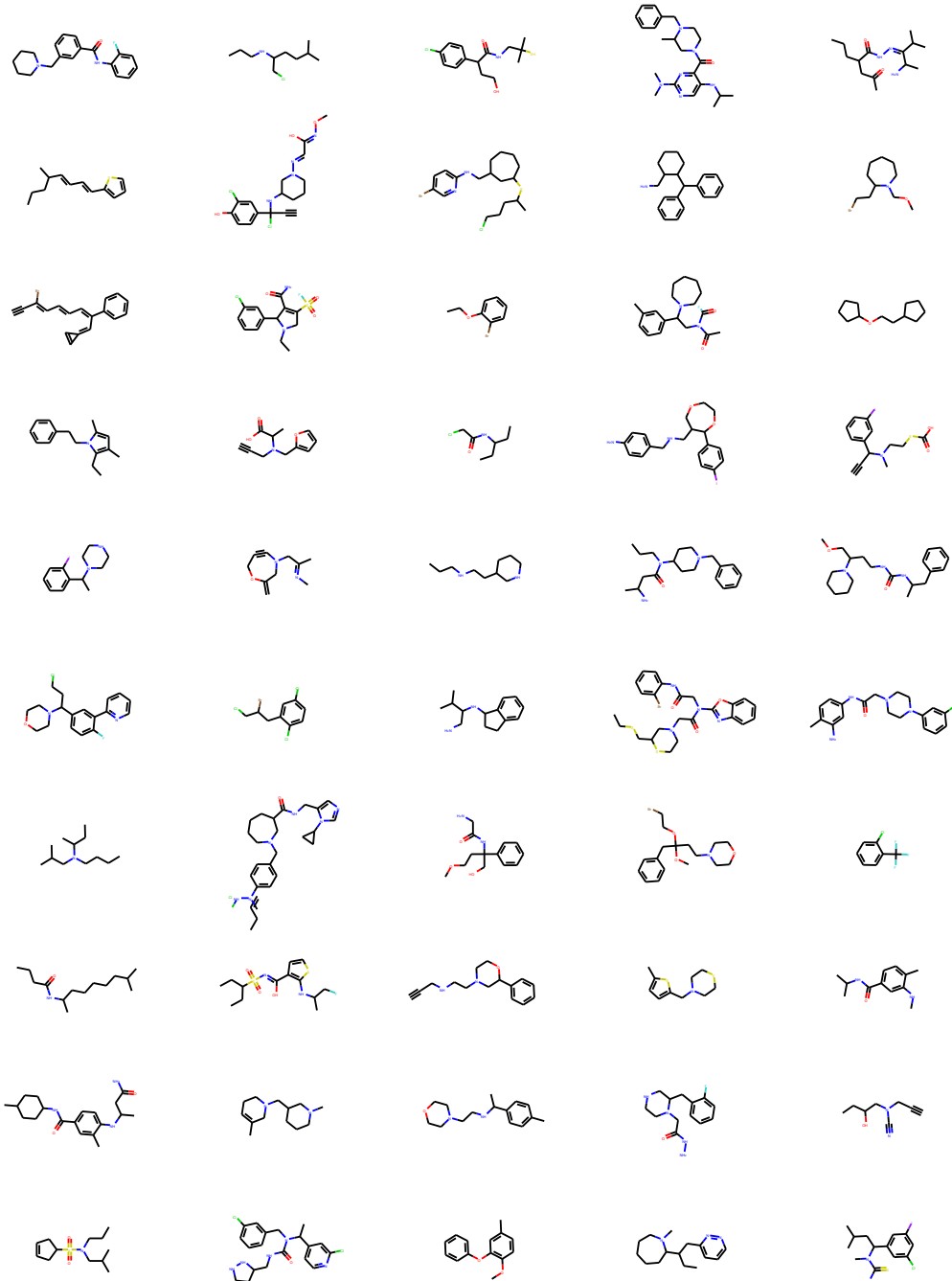

Figure 5: 50 molecules sampled from prior.

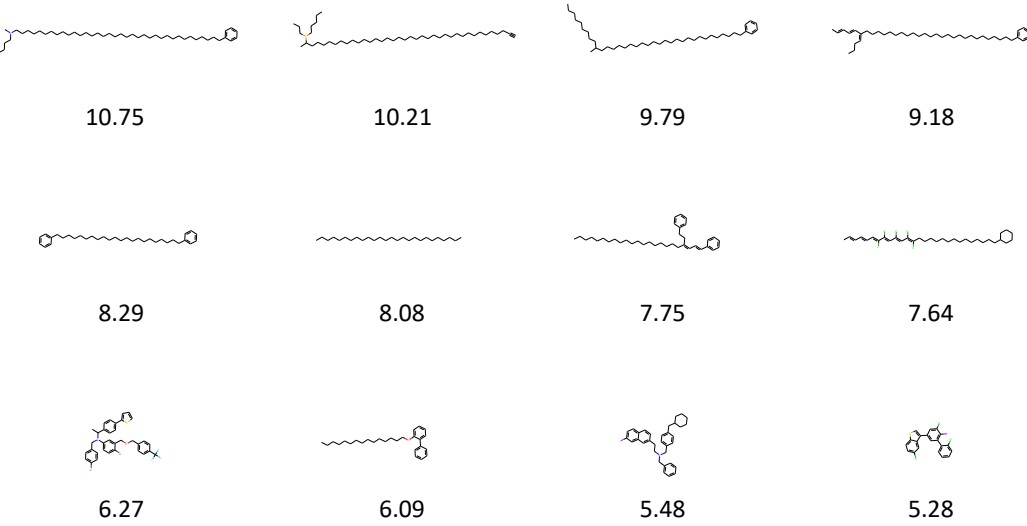

Figure 6: Molecule samples with high penalized logp score generated by GraphAF.

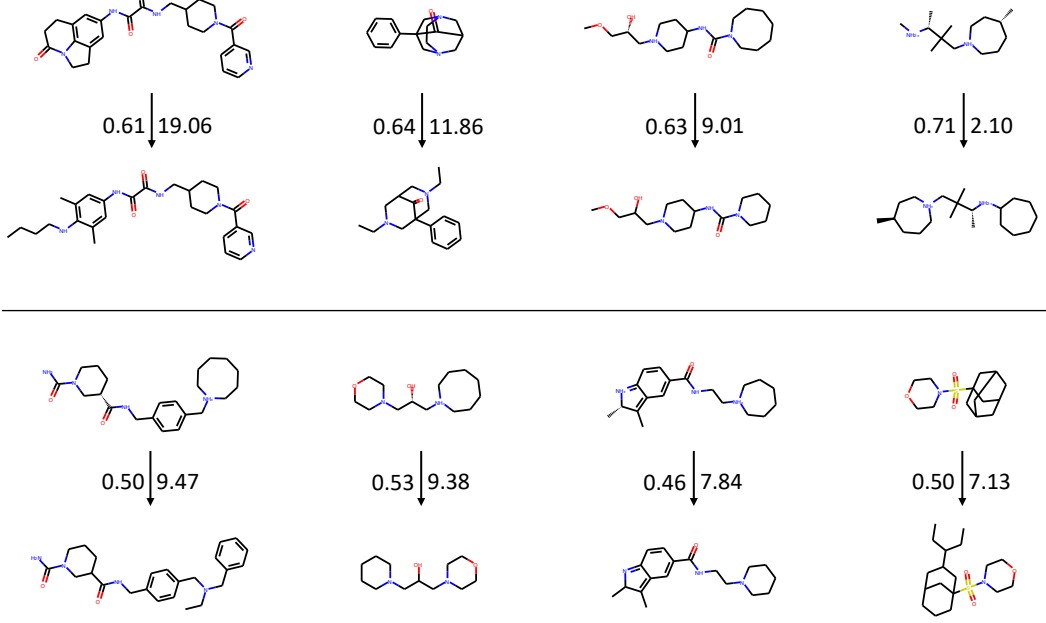

Figure 7: More results on constrained property optimization for penalized logP score. Numbers beside the arrow denote similarity and improvement of the given molecule pair respectively.

