# OpenReview forum: "GraphAF: a Flow-based Autoregressive Model for Molecular Graph Generation"
_ICLR.cc/2020/Conference — Accept (Poster)_

### Official Review · AnonReviewer1 · 2019-10-07
**Official Blind Review #1**

**Rating:** 6

**Review:**

This paper proposes a generative model architecture for molecular graph generation based on autoregressive flows. The main contribution of this paper is to combine existing techniques (auto-regressive BFS-ordered generation of graphs, normalizing flows, dequantization by Gaussian noise, fine-tuning based on reinforcement learning for molecular property optimization, and validity constrained sampling). Most of these techniques are well-established either for data generation with normalizing flows or for molecular graph generation and the novelty lies in the combination of these building blocks into a framework. Training can be carried out in parallel over the sequential generation process, as no hidden states with sequential dependency are assumed (unlike a regular RNN). Experimental validation is carried out on a standard ZINC molecule generation benchmark (graphs with up to 48 nodes) and the reported metrics are competitive with recent related work.

Overall, the paper is very well written, nicely structured and addresses an important problem. The framework in its entirety is novel, but the building blocks of the proposed framework are established in prior work and the idea of using normalizing flows for graph generation has been proposed in earlier work (see [1] and [2]). Nonetheless, I find the paper relevant for an ICLR audience and the quality of execution and presentation of the paper is good.

I have two major (technical) concerns with the flow-based formulation used in the paper with regards to order-invariance and the utilized de-quantization scheme.
* Order-invariance: The paper states that the “exact density of each molecule can be efficiently computed by the change-of-variables formula”. This seems to be incorrect, as the exact density is a product over all order-specific densities for all possible permutations in which the molecular graph can be represented. The change-of-variables formula does not provide an efficient way to circumvent this order-invariance issue, at least not in the way it is presented in the paper. Even when using BFS-ordered representations, the subspace of possible permutations is still typically too large to allow for efficient evaluation of the exact density. I suspect that the authors assume a canonical ordering of the graph representations, which is a strong assumption, but does not seem to be mentioned in the paper. How is the canonical ordering chosen? How is local structural symmetry broken in a consistent manner?
* De-quantization: The de-quantization scheme used in this paper seems to be ill-suited for categorical variables. What motivates the use of adding Gaussian noise to categorical (one-hot encoded) variables, other than that it seems to work OK in the reported experiments? Adding Gaussian noise in this way can move these variables outside of the probability simplex — is this a valid technique in the framework of normalizing flows? Adding Gaussian noise makes sense if the data represents quantized continuous data, e.g. bit-quantized image data, but I have concerns about the validity of using this method for categorical data (both edge type and node features are categorical in this application). Other comparable generative models for graph-structured data use a relaxed discrete distribution (concrete / Gumbel softmax), e.g. in MolGAN [De Cao & Kipf (2018)], to address this issue — would this also be applicable here?

I think that these two issues will have to be addressed before this paper can be considered for publication, and I recommend a weak reject at this point.

[1] Madhawa et al., GraphNVP: An invertible flow model for generating molecular graphs. (2019)
[2] Liu et al., Graph Normalizing Flows. (2019) — not cited


UPDATE:

My two main technical concerns have been addressed in the rebuttal and I think that the revised version of the paper can be accepted to ICLR (my comment w.r.t. novelty still holds and hence I recommend 'weak accept').

**Experience Assessment:**

I have published one or two papers in this area.

**Review Assessment: Checking Correctness Of Derivations And Theory:**

I assessed the sensibility of the derivations and theory.

**Review Assessment: Checking Correctness Of Experiments:**

I assessed the sensibility of the experiments.

**Review Assessment: Thoroughness In Paper Reading:**

I made a quick assessment of this paper.

---

> ### Author Response · Authors · 2019-11-12
> **Response to the AnonReviewer1 cont.**
>
>
> Q3: * De-quantization: The de-quantization scheme used in this paper seems to be ill-suited for categorical variables. What motivates the use of adding Gaussian noise to categorical (one-hot encoded) variables, other than that it seems to work OK in the reported experiments? Adding Gaussian noise in this way can move these variables outside of the probability simplex — is this a valid technique in the framework of normalizing flows? Adding Gaussian noise makes sense if the data represents quantized continuous data, e.g. bit-quantized image data, but I have concerns about the validity of using this method for categorical data (both edge type and node features are categorical in this application). Other comparable generative models for graph-structured data use a relaxed discrete distribution (concrete / Gumbel softmax), e.g. in MolGAN[3], to address this issue — would this also be applicable here?
> A3: Actually, instead of Gaussian noise, we used the uniform noise (Equation 5) for de-quantization.
> The same techniques have also been used in other normalizing flow methods for discrete data  (e.g. GraphNVP[3], RealNVP[5], Glow[6]) and also shown very effective.
>
> Note that Gumbel softmax and dequantization are techniques for two very different problems on discrete data. The former one is used to backpropagate the gradient through discrete variables, while dequantization is used to transform discrete data into continuous since the invertible mappings defined by normalizing flows are mainly for continuous data.
>
> We hope the above response could address your concerns. Please let us know if you have other questions. We’re happy to further answer.
>
>
> [1] Liu et al., Graph Normalizing Flows. arXiv 2019.05.
> [2] You et al. GraphRNN: Generating Realistic Graphs with Deep Auto-regressive Models. ICML 2018.
> [3] Madhawa et al., GraphNVP: An invertible flow model for generating molecular graphs. arXiv 2019.05.
> [4] Popova et al. Molecularrnn: Generating realistic molecular graphs with optimized properties. arXiv preprint arXiv:1905.13372, 2019.
> [5] Dinh et al., Density Estimation using Real NVP. ICLR’17.
> [6] Diederik P. Kingma, Prafulla Dhariwal. Glow: Generative Flow with Invertible 1×1 Convolutions. NIPS’18.

---

> > ### Comment · AnonReviewer1 · 2019-11-13
> > **Response (part 2)**
> >
> > Thank you for your response.
> >
> > I have another clarification question regarding A3:
> >
> > As mentioned in my initial review, RealNVP and Glow use their de-quantization technique on bit-quantized image data (unless I misunderstand their technique) where simply adding noise is justified as it "spreads out" the discrete values. For binary or categorical data, however, I still think that this technique is problematic, as my point from my original review (adding noise can move points outside of the probability simplex) still holds even if you use uniform noise instead of Gaussian noise (apologies for this misunderstanding). GraphNVP seems to have a similar issue (this paper is currently under review at ICLR as well and one reviewer pointed out the same concern), so I think it is not valid to point to them for justification of this approach.
> >
> > Would you be able to further clarify or justify your method?

---

> > > ### Author Response · Authors · 2019-11-14
> > > **Response to the AnonReviewer1**
> > >
> > > Thank you for your quick response! The discussion is really helpful. Looking forward to your reply.  The answers to your concerns are listed below.
> > >
> > > Q3: As mentioned in my initial review, RealNVP and Glow use their de-quantization technique on bit-quantized image data (unless I misunderstand their technique) where simply adding noise is justified as it "spreads out" the discrete values. For binary or categorical data, however, I still think that this technique is problematic, as my point from my original review (adding noise can move points outside of the probability simplex) still holds even if you use uniform noise instead of Gaussian noise (apologies for this misunderstanding). GraphNVP seems to have a similar issue (this paper is currently under review at ICLR as well and one reviewer pointed out the same concern), so I think it is not valid to point to them for justification of this approach.
> > >
> > > A3:  Moving the points outside of the probability simplex actually does not matter as the normalizing flows are actually defined on general continuous data. The dequantization techniques allow mapping the discrete data into continuous one by adding a small noise to each dimension. By adding a noise from U[0,1), we can assure that the range of different categories will not overlap. For example, after dequantization, the value of 1-entry in the one-hot vector lie in [1,2) while the 0-entry lie in [0,1). We can also map the generated continuous data back to the discrete data by using the argmax function in the generation process.
> > > Theoretically, as shown in [1] (eq3-6) and [2],  training a continuous density model on uniform dequantized data can be interpreted as maximizing a lower bound on the log-likelihood for the original discrete data. This statement holds for both image data and binary/categorical data mathematically.  In addition, as suggested in [2], instead of adding random uniform noise to each discrete data for dequantization, a more advanced dequantization technique is to treat the noise as hidden variables and use variational inference to infer the optimum noise added to each discrete data. We will explore this in our future work.
> > >
> > > We have added a section to discuss the dequantization techniques in the appendix. We are happy to further discuss this if you still have questions.
> > >
> > > [1] Theis, Lucas, Aäron van den Oord, and Matthias Bethge. "A note on the evaluation of generative models." arXiv preprint arXiv:1511.01844 (2015).
> > > [2] Ho, Jonathan, et al. "Flow++: Improving flow-based generative models with variational dequantization and architecture design." arXiv preprint arXiv:1902.00275 (2019).

---

> > > > ### Comment · AnonReviewer1 · 2019-11-14
> > > > **Response**
> > > >
> > > > Thanks a lot, this clarification really helps and my initial concern no longer holds. Thank you also for providing these two additional references and for devoting a section in your appendix to this issue. I think with these changes I feel comfortable in recommending acceptance of this paper.

---

> > > > > ### Author Response · Authors · 2019-11-14
> > > > > **Response**
> > > > >
> > > > > Thanks for your quick response! Your reviews really help improve the paper. We really appreciate it.

---

> ### Author Response · Authors · 2019-11-12
> **Response to the AnonReviewer1**
>
> Thanks for your comments and suggestions. The response to some of your concerns are listed below:
> Q1: The framework in its entirety is novel, but the building blocks of the proposed framework are established in prior work and the idea of using normalizing flows for graph generation has been proposed in earlier work (see GraphNVP[4] and GNF). Nonetheless, I find the paper relevant for an ICLR audience and the quality of execution and presentation of the paper is good.
> A1: Our work is indeed related to the two work Graph Normalizing Flows[1] (GNF, we have added the missing reference) and GraphNVP[3]. However, our work is fundamentally different from their work. GNF defines a normalizing flow from a base distribution to the hidden node representations of a pretrained Graph Autoencoders.  The graph generation is done through two separate stages by first generating the node embeddings with the normalizing flows and then generate the graphs based on the generated node embeddings in the first stage. In GraphAF, we define an autoregressive flow from a base distribution to the molecular graph structures, which can be trained end-to-end. GraphNVP also defines a normalizing flow from a base distribution to the molecular graph structures. However, the generation process of GraphNVP is one-shot, which cannot effectively capture graph structures and also cannot guarantee the validity of generated molecules (only 40% validity rate). In our GraphAF, we formulate the generation process as a sequential decision process and effectively capture the subgraph structures based on graph neural networks, based on which we define a policy function to generate the nodes and edges. The sequential generation process also allows to incorporate the chemical rules. As a result, the validity of the generated molecules can be guaranteed (100% validity rate). Moreover, we can effectively optimize the properties of generated molecules by fine-tuning the policy with reinforcement learning, which is not feasible in GraphNVP.
>
> Q2: Order-invariance: The paper states that the “exact density of each molecule can be efficiently computed by the change-of-variables formula”. This seems to be incorrect, as the exact density is a product overall order-specific densities for all possible permutations in which the molecular graph can be represented. The change-of-variables formula does not provide an efficient way to circumvent this order-invariance issue, at least not in the way it is presented in the paper. Even when using BFS-ordered representations, the subspace of possible permutations is still typically too large to allow for efficient evaluation of the exact density. I suspect that the authors assume a canonical ordering of the graph representations, which is a strong assumption, but does not seem to be mentioned in the paper. How is the canonical ordering chosen? How is local structural symmetry broken in a consistent manner?
> A2: Thanks for pointing this out!! Following existing work on graph generation—GraphRNN[2] and MolecularRNN[4]—we use the BFS-ordering of a graph (mentioned in Section 4.2). BFS-ordering has been shown very effective for graph generation in previous work, which can effectively limit the number of edge predictions made for each node and hence significantly accelerate training [2].
>
> And you’re right on that we cannot calculate the exact likelihood of a molecule, which requires calculating all the permutation of the molecule. What we mean is that we can calculate the exact likelihood of the canonical order (BFS-order) of a molecule. We’ve already revised this in the new version.

---

> > ### Comment · AnonReviewer1 · 2019-11-13
> > **Response (part 1)**
> >
> > Thank you for your response.
> >
> > I have two follow-up questions to part 1 of your response to avoid potential misunderstandings.
> >
> > A1: You mention that GraphNVP is not compatible with a reinforcement learning objective for fine tuning. Is this because of the one-shot nature of the generation process that you refer to? One example that comes to my mind where a one-shot generative process is combined with an RL objective is MolGAN [1] -- maybe you can comment on this.
> >
> > [1] De Cao & Kipf, "MolGAN: An implicit generative model for small molecular graphs", 2018
> >
> > A2: How do you canonically order nodes *within* the BFS front of the BFS-ordering? To me it seems like typical BFS-ordering only gives you a partial ordering of the nodes in a graph as nodes within the BFS front are still ordered arbitrarily. Hence you would still not have the exact likelihood unless you find a way to break the symmetry consistently within the BFS front. Please correct me if I'm wrong.

---

> > > ### Author Response · Authors · 2019-11-14
> > > **Response to the AnonReviewer1**
> > >
> > > Thank you for your quick response! The discussion is really helpful. Looking forward to your reply.  The answers to your concerns are listed below.
> > >
> > > Q1: You mention that GraphNVP is not compatible with a reinforcement learning objective for fine-tuning. Is this because of the one-shot nature of the generation process that you refer to? One example that comes to my mind where a one-shot generative process is combined with an RL objective is MolGAN [1] -- maybe you can comment on this.
> > >
> > > A1: Very good point! Indeed, GraphNVP is compatible with the objective used in MolGAN. However, note that the approach used in MolGAN is not based on RL and is actually based on the one-step policy gradient algorithm, which is not the classical RL. The classical RL problem is a sequential decision process, which involves a series of states and actions. This process also allows us to introduce intermediate rewards (e.g., the penalization for valency check in each step defined in Section 4.4) and final rewards. For the approach used in MolGAN---which is only one-step decision---we are only able to provide final rewards but not able to leverage the intermediate rewards (e.g., the chemical rules for valency check).
> > >
> > > To avoid misunderstanding, we have removed this statement in section 2.
> > >
> > >
> > > Q2: How do you canonically order nodes *within* the BFS front of the BFS-ordering? To me, it seems like typical BFS-ordering only gives you a partial ordering of the nodes in a graph as nodes within the BFS front are still ordered arbitrarily. Hence you would still not have the exact likelihood unless you find a way to break the symmetry consistently within the BFS front. Please correct me if I'm wrong.
> > >
> > > A2: Thanks for raising this point again. Yes, you are right that nodes within the BFS front are still ordered arbitrarily.  GraphAF is trained on all possible BFS orderings. This can be done by first randomly permuting the adjacency matrix and then randomly pick a node as BFS front. By canonical order(BFS-order), we meant that all the orders of graphs we used to train GraphAF are BFS orders. We meant the exact density of each molecule under a given order (which is sampled each time that  we load a batch of training graphs) can be efficiently computed by the change-of-variables formula. We have revised the section 4.2 because we think the word “canonical” is a little bit misleading. Thanks for the suggestions!!

---

> > > > ### Comment · AnonReviewer1 · 2019-11-14
> > > > **Response**
> > > >
> > > > Thanks a lot for your detailed clarifications regarding my questions on the RL objective and the BFS ordering. This is indeed clear now and I think that the revised statements in the paper should avoid further confusion on this point.

---

### Official Review · AnonReviewer2 · 2019-10-23
**Official Blind Review #2**

**Rating:** 6

**Review:**

This paper proposes a new invertible-flow based graph generation model.
The main difference from the previous flow-based generation model (GraphNVP) is the choice of flow mappings (coupling flow --> autoregressive flow).
The authors formulate conditional probabilities of iterative node/edge generation. Iterative sampling scheme naturally allows incorporating validity checks in each sampling step, assuring 100% validity in graph generation. The paper also proposes an implementation of molecule lead optimization combined with a RL framework. The experimental results show superiority in terms of valid graph generation and property optimization.


Overall, the paper is written well. I feel no difficulty in understanding the main idea and the equations in the paper.

Introduction of the normalizing flows (Sec 3.1) can be expanded to reach non-expert users. Advantages of using invertible flows (against other generative models such as GANs and VAEs) are not described rigorously in the current manuscript. I also suggest citing a nice review for invertible flows appeared recently:

Ivan Kobyzev, Simon Prince, and Marcus A Brubaker, ``Normalizing Flows: Introduction and Ideas'', arXiv: 1908.09257, 2019.


Explanations of the Sec 4.4 (+ appendix B) is simply insufficient to reproduce the experiments. More descriptions or references are required.


Experimental results seem promising. A better validity score than the previous flow model illustrates the efficacy of the autoregressive flow against the coupling flow.
The performance on the property optimization (Table 3) seems brilliant. However, there is no discussion why the combination of the autoregressive flow and the RL performs greatly, compared to baselines. Some discussions will help the community to further improve the optimization tasks in the future.

+ Overall, a good paper. well written, easy to understand.
+ A new variant of the invertible-flow based graph generation model.  The novelty lies in the iterative generation process, naturally combined with the autoregressive flow.
+ Superior to the one-shot flow baseline (GraphNVP) even if additional validity checks are omitted (Table 2)
+ Good performances in property optimizations (Table 3, 4)
- The explanation for RL process is simply insufficient for reproduction.
-- No discussions about reasons why GraphAF+RL performs great in property optimization.


**Experience Assessment:**

I have published one or two papers in this area.

**Review Assessment: Checking Correctness Of Derivations And Theory:**

I assessed the sensibility of the derivations and theory.

**Review Assessment: Checking Correctness Of Experiments:**

I assessed the sensibility of the experiments.

**Review Assessment: Thoroughness In Paper Reading:**

I read the paper at least twice and used my best judgement in assessing the paper.

---

> ### Author Response · Authors · 2019-11-12
> **Response to the AnonReviewer2**
>
> Thanks for your comments and suggestions. The response to your concerns are listed below:
>
> Q1: The introduction of the normalizing flows (Sec 3.1) can be expanded to reach non-expert users. Advantages of using invertible flows (against other generative models such as GANs and VAEs) are not described rigorously in the current manuscript. I also suggest citing a nice review for invertible flows appeared recently.
> Explanations of the Sec 4.4 (+ appendix B) is simply insufficient to reproduce the experiments. More descriptions or references are required.
> A1: Thank you for suggestions. We’ve already revised this section and also cited a new introduction and survey paper on normalizing flows [1]. Advantages of flow are briefly introduced in the introduction. We’ve also revised and extended Sec 4.4 in the revised version.
>
> Q2: No discussion why the combination of the autoregressive flow and the RL performs greatly, compared to baselines. Some discussions will help the community to further improve the optimization tasks in the future.
> A2: This is a very good point!! As defined in Sec 4.4, our RL process is close to the one in previous work GCPN[2]. Therefore, the good property optimization performance is believed to come from the flexibility of flow. Compared with the GAN model used in GCPN[2], which is known to suffer from the mode collapse problem, flow is flexible at modeling complex distributions and generating diverse data (as shown in Table 2). This allows to explore a variety of molecule structures in the RL process for molecule properties optimization.
>
> We hope the above response could address your concerns.
>
> [1] Ivan Kobyzev, Simon Prince, Marcus A. Brubaker. Normalizing Flows: Introduction and Ideas. arXiv:1908.09257.
> [2] You et al. Graph Convolutional Policy Network for Goal-Directed Molecular Graph Generation. NeurIPS 2018.

---

> > ### Comment · AnonReviewer2 · 2019-11-12
> > **Cannot confirm the manuscript update**
> >
> > Hi,
> >
> > Thank you for your answers.
> >
> > Unfortunately I cannot confirm the updated manuscript.
> > In fact, the header info of this page says: 26 Sep 2019 (modified: 26 Sep 2019) so not updated.
> >
> > Bests

---

> > > ### Author Response · Authors · 2019-11-12
> > > **Response to the AnonReviewer2**
> > >
> > > Hi,
> > > We have just uploaded the paper. I believe it is all set now.
> > >
> > > Bests

---

> > > > ### Comment · AnonReviewer2 · 2019-11-14
> > > > **Thanks**
> > > >
> > > > I can read the updated manuscript now. Thanks!

---

### Official Review · AnonReviewer3 · 2019-10-26
**Official Blind Review #3**

**Rating:** 6

**Review:**

# Post Rebuttal

The authors have partially and satisfactorily addressed my concerns. In line of this I am raising my score to Weak Accept.

This paper proposes a new molecular graph generative model (GraphAF) which fuses the best of two worlds of generative networks - reversible flow and autoregressive mode. Such integration enjoys a) faster training due to parallel computation b) molecular validity checker during inference supported by sequential sampling process and c) exact likelihood maximisation due to invertible encoder. In lieu of such advantages, the model trains two times faster than the existing state-of-the-art and generates 100% valid molecules when trained on ZINC dataset. Further, it also demonstrates that additionally if the chemical properties are optimised during training with reinforcement learning policy then GraphAF outperforms all the prior works.

Although the paper presents an interesting fusion of different generative models, in its current form it leans towards rejection due to the following factors:
1) The empirical validation of GraphAF is contained to single dataset - ZINC with a maximum of 38 atoms. From the table 2, it seems to me every prior method works pretty well on important metrics. There is very little room for improvement. I recommend including results on QM9 and CEPDB datasets.
2) The model being data-agnostic, it makes sense to evaluate them on generic graph datasets - synthetic and real.
3) The novelty of the model is limited. The flow-based graph generative model is introduced in Graph Normalizing Flow (GNF) (NeurIPS'19, NeurIPS'18 workshop). The reversible flow is extended to whole graph in GraphNVP. Unlike GNF, GraphNVP and GraphAF do away with decoder. The major difference being the sampling process - one-shot to sequential.
I am willing to improve my rating given that some of this points are addressed.

Clarification:
1. What are the inputs edge-mlp's operate on ? Given the generation step is sequential, it is not clear to me why all the node embeddings H_i^L is given as input in eq (8). I also noted that the dimension of H_i^L varies with size of sub-graphs. Also note mismatch in the notation 'f' used in algorithm 1 and 'g' from the main text.
2. Please compare inference time.

Other weakness:
1. Due to invertible flow modeling, the latent space is usually restricted to small dimension. In current case it is 9 for node feature and 3 for edge features. This drawback alongside the sequential edge generation prevents GraphAF from scaling to complex and large graphs with many labels.
2. Moreover, GraphAF utilizes only single layer of flow i.e., eq (9). This is clearly not sufficient to model complex graphs. And in its current form it is not clear how one can extend to multi-layer flow.
3. The encoder modeling in GraphAF also shares similarity with Variational graph auto-encoder. Instead of constraining latent distribution using KL divergence, GraphAF maximizes graph likelihood to enforce base distribution.


**Experience Assessment:**

I have read many papers in this area.

**Review Assessment: Checking Correctness Of Derivations And Theory:**

I carefully checked the derivations and theory.

**Review Assessment: Checking Correctness Of Experiments:**

I carefully checked the experiments.

**Review Assessment: Thoroughness In Paper Reading:**

I read the paper thoroughly.

---

> ### Author Response · Authors · 2019-11-12
> **Response to the AnonReviewer3 cont.**
>
>
> Q3: The encoder modeling in GraphAF also shares similarities with Variational graph auto-encoder. Instead of constraining latent distribution using KL divergence, GraphAF maximizes graph likelihood to enforce base distribution.
> A3: In general, normalizing flows are indeed related to variational auto-encoders, both of which tried to explicitly model the data density and aim to maximize the data likelihood. However, flow-based methods are fundamentally different from VAE in the following perspectives: (1) flow-based methods define an invertible mapping between the latent space and observation space; (2) flow-based methods allow to calculate the exact likelihood while VAE methods can only optimize a lower bound.
>
> We hope the above responses address your concerns. Please let us know if you have other questions. We’re happy to further answer the questions.
>
> [1] You et al. Graph Convolutional Policy Network for Goal-Directed Molecular Graph Generation. NeurIPS 2018.
> [2] Jin et al. Junction Tree Variational Autoencoder for Molecular Graph Generation. ICML 2018.
> [3] Liu et al., Graph Normalizing Flows. arXiv 2019.05.
> [4] You et al. GraphRNN: Generating Realistic Graphs with Deep Auto-regressive Models. ICML 2018.
> [5] Madhawa et al., GraphNVP: An invertible flow model for generating molecular graphs. arXiv 2019.05.
> [6] Zachary M. Ziegler,  Alexander M. Rush. Latent Normalizing Flows for Discrete Sequences. ICML’19.

---

> ### Author Response · Authors · 2019-11-12
> **Response to the AnonReviewer3  cont.**
>
>
> Q3: The novelty of the model is limited. The flow-based graph generative model is introduced in Graph Normalizing Flow (GNF) (NeurIPS'19, NeurIPS'18 workshop). The reversible flow is extended to whole graph in GraphNVP. Unlike GNF, GraphNVP[4] and GraphAF do away with decoder. The major difference being the sampling process - one-shot to sequential.
> A3: Thanks for pointing out the related work. Our work is indeed related to the two work Graph Normalizing Flows (GNF) and GraphNVP[5]. However, our work is fundamentally different from the two works. GNF defines a normalizing flow from a base distribution to the hidden node representations of a pre-trained Graph Autoencoders.  The graph generation is done through two separate stages by first generating the node embeddings with the normalizing flows and then generate the graphs based on the generated node embeddings in the first stage. In GraphAF, we define an autoregressive flow from a base distribution to the molecular graph structures, which can be trained end-to-end. GraphNVP also defines a normalizing flow from a base distribution to the molecular graph structures. However, the generation process of GraphNVP is one-shot, which cannot effectively capture graph structures and also cannot guarantee the validity of generated molecules (only 40% validity rate). In our GraphAF, we formulate the generation process as a sequential decision process and effectively capture the subgraph structures based on graph neural networks, based on which we define a policy function to generate the nodes and edges. The sequential generation process also allows incorporating the chemical rules. As a result, the validity of the generated molecules can be guaranteed (100% validity rate). Moreover, we can effectively optimize the properties of generated molecules by fine tuning the policy with reinforcement learning, which is not feasible in GraphNVP.
>
>
> Clarification:
> Q1: What are the inputs edge-mlp's operate on? Given the generation step is sequential, it is not clear to me why all the node embeddings H_i^L is given as input in eq (8). I also noted that the dimension of H_i^L varies with size of sub-graphs. Also note mismatch in the notation 'f' used in algorithm 1 and 'g' from the main text.
> A1: H_i^L is the node embeddings of the current sub-graphs in the i-th step. The inputs of Edge-MLP’s include the graph-level embedding of the current sub-graphs h_i, the embedding of new node X_i, and the embedding of the previous j-th node X_j. We have added these details of Eq. (8) in the revised version.
> Thanks for pointing this out. We’ve already corrected the notations in Algorithm 1.
>
>
> Q2: Please compare inference time.
> A2: Let {V, E} be the set of nodes and edges in a graph. The inference time of our GraphAF is O(|V|+ c|V|), where c is the maximum number of link predictions for each node in BFS order.
>
> Other weakness:
> Q1. Due to invertible flow modeling, the latent space is usually restricted to small dimension. In current case it is 9 for node feature and 3 for edge features. This drawback alongside the sequential edge generation prevents GraphAF from scaling to complex and large graphs with many labels.
> A1: A good point! With BFS order, the complexity of GraphAF scales linearly to the number of nodes or edges. Therefore, the size of the graph is not an issue. To scale to graphs with many node types, we can represent each node type with a low-dimensional vector instead of with a one-hot high-dimensional vector. Similar ideas have already been explored in using normalizing flows for text generation [6] (the size of vocabulary is very large).
>
> Q2. Moreover, GraphAF utilizes only single layer of flow i.e., eq (9). This is clearly not sufficient to model complex graphs. And in its current form it is not clear how one can extend to multi-layer flow.
> A2:  We only use one single layer of flow since it has already been shown very powerful for modeling molecular graph structures in different data sets. However, the framework is very general and can be easily extended to multi-layer flow for modeling complex graphs. Specifically, we can construct a T-layer flow to map molecular graph structures to base distribution (z->\epsilon_T->\epsilon_{T-1}->...->\epsilon_1). In the t-th layer, we take \epsilon_t as the features of nodes and edges. Since \epsilon_t is continuous, we can directly input it into R-GCN (see Eq.(8)) without the dequantization process, and then perform the transformation to \epsilon_{t-1} as defined in Eq.(10) .

---

> ### Author Response · Authors · 2019-11-12
> **Response to the AnonReviewer3**
>
> Thank you very much for your constructive comments! We have conducted more experiments according to your suggestions. The response to some of your questions are listed below:
>
> Q1: The empirical validation of GraphAF is contained to the single dataset - ZINC with a maximum of 38 atoms. From table 2, it seems to me every prior method works pretty well on important metrics. There is very little room for improvement. I recommend including results on QM9 and CEPDB datasets.
> A1: Thanks for your suggestion on more datasets. The reason that we only use ZINC data set is that we followed the experiment setting in existing work including GCPN[1] and JT-VAE [2]. According to your suggestions, we conducted additional experiments on QM9 (134K molecules in total) and MOSES (1.9M molecules in total). We didn’t use CEPDB since currently we can’t access to this dataset. The results on QM9 and MOSES data set are summarized below:
> |  Data.  | Valid     | Valid w/o check  | Uniqueness | Novelty | Reconstruction |
> | QM9     |    100    |    67                     |       94.51      |    88.83  |       100               |
> |MOSES  |     100   |     71                   |       99.99      |    100     |       100                |
> We can see that our method can generate valid, unique, and novel molecules with different training data sets.
>
> On the task of density modeling, we agree that different methods perform comparably. However, noting that GraphAF (1) can achieve 68% validity rate even without leveraging chemical domain knowledge thanks to the strong capacity of normalizing flow framework (GCPN can only reach 20%); (2) enjoys parallel training and is therefore much more efficient than existing methods. Moreover,  on the more challenging and more important tasks of drug discovery—Property Optimization and Constrained Property Optimization—GraphAF achieves the state-of-the-art performance (Table 5 and Table 6).
>
>
> Q2: The model being data-agnostic, it makes sense to evaluate them on generic graph datasets - synthetic and real.
> A2: Note that GraphAF is mainly designed for molecular graph generation. However, it is indeed very general and can be generalized to generate different types of graphs by changing the Edge-MLPs and Node-MLP functions (Equation 8).  Specifically, we follow the experiment setup of GNF[3](Sec.5.2 in the original paper) and run GraphAF on two generic graph datasets: COMMUNITY-SMALL(synthetic) , EGO-SMALL(real). The results are as follows:
> Community Small  					Ego Small
> Degree     Cluster     Orbit 	Degree       Cluster      Orbit
> ————————————————————————————————
> GraphRNN   0.08           0.12       0.04		  0.09           0.22         0.003
> GNF		0.20	       0.20      0.11		  0.03	          0.10         0.001
> GraphAF      0.18            0.20      0.02		 0.03 	          0.11	    0.001
> ————————————————————————————————
> GraphRNN   0.03           0.01       0.01		  0.04           0.05         0.06
> GNF		0.12	       0.15      0.02		  0.01	          0.03        0.0008
> GraphAF      0.06            0.10      0.015		 0.04 	          0.04 	    0.008
>
> The above results demonstrate that GraphAF, when applied to generic graphs, can consistently yield comparable or better results compared with existing state-of-the-art approaches GNF[3] and GraphRNN[4]. We have added these results in the revised version.

---

### Author Response · Authors · 2019-11-12
**Response to all the reviewers and area chair**

We would like first to thank all the reviewers for your constructive reviews. We’ve revised the paper according to your reviews. Specifically, we have made the following changes:
1. We conduct additional experiments on another two molecule data sets QM9 and MOSES and two generic graph data sets in Section 5.2. The results are available at Table 3, 4. Results show that our proposed method GraphAF can still get state-of-the-art or competitive results on these data sets. We also present some generated examples of generic graph in the appendix.
2. We expand the description of normalizing flows to make the paper more self-contained. A citation of the survey paper on normalizing flows is also given for reference. We also explained the RL process in Section 4.4 in more details.
3. We discuss the difference between our work and existing work including Graph Normalizing Flows and GraphNVP in more detail in Section 2.
4. We give a detailed explanation on why GraphAF + RL pipeline works well on the tasks of property optimization in Section 5.2.
5. We revise the statement of “calculating the exact density of each molecule” with GraphAF to “calculating the exact density of each molecule under a given order”.
6. We added a section to discuss the dequantization techniques in the appendix.

---

### Public Comment · ~Ziqi_Chen1 · 2020-07-27
**Questions about "Constrained Property Optimization" experiment**

Hi,

I noticed that in your experiment about constrained property optimization, your test set contains 800 molecules with the lowest penalized logP in ZINC250k. I think this test set is not the same with the test set used in JT-VAE and GCPN. Actually, the description about test set in JT-VAE is:

"To provide the greatest challenge, we selected 800 molecules with the lowest property score y(·) from the test set."

The test set in the above sentence represents the test set used in [1].

I checked the test set provided in the google drive, and found that the penalized logp values of compounds in your test set range from -62.5 to -8.2. Considering that the penalized logp values in the test set of JT-VAE and GCPN range from -11.0 to -0.5, I think it would be unfair to compare the results on this test set with the results of JT-VAE and GCPN.

[1]: Kusner, Matt J., Brooks Paige, and José Miguel Hernández-Lobato. "Grammar variational autoencoder." arXiv preprint arXiv:1703.01925 (2017).

---

> ### Author Response · Authors · 2020-07-28
> **Thanks for pointing it out.**
>
> Hi, Ziqi,
> We are actually aware of this issue and will report the new results as soon as possible.

---

### Decision · Program_Chairs · 2019-12-19

**Decision:**

Accept (Poster)

**Comment:**

All reviewers agreed that this paper is essentially a combination of existing ideas, making it a bit incremental, but is well-executed and a good contribution.  Specifically, to quote R1:

"This paper proposes a generative model architecture for molecular graph generation based on autoregressive flows. The main contribution of this paper is to combine existing techniques (auto-regressive BFS-ordered generation of graphs, normalizing flows, dequantization by Gaussian noise, fine-tuning based on reinforcement learning for molecular property optimization, and validity constrained sampling). Most of these techniques are well-established either for data generation with normalizing flows or for molecular graph generation and the novelty lies in the combination of these building blocks into a framework. ... Overall, the paper is very well written, nicely structured and addresses an important problem. The framework in its entirety is novel, but the building blocks of the proposed framework are established in prior work and the idea of using normalizing flows for graph generation has been proposed in earlier work. Nonetheless, I find the paper relevant for an ICLR audience and the quality of execution and presentation of the paper is good."